# Molecular Profiles of Serum-Derived Extracellular Vesicles in High-Grade Serous Ovarian Cancer

**DOI:** 10.3390/cancers14153589

**Published:** 2022-07-23

**Authors:** Li Zhao, Sara Corvigno, Shaolin Ma, Joseph Celestino, Nicole D. Fleming, Richard A. Hajek, Adrian Lankenau Ahumada, Nicholas B. Jennings, Erika J. Thompson, Hongli Tang, Shannon N. Westin, Amir A. Jazaeri, Jianhua Zhang, P. Andrew Futreal, Anil K. Sood, Sanghoon Lee

**Affiliations:** 1Department of Genomic Medicine, The University of Texas MD Anderson Cancer Center, Houston, TX 77030, USA; lzhao7@mdanderson.org (L.Z.); jzhang22@mdanderson.org (J.Z.); afutreal@mdanderson.org (P.A.F.); 2Department of Gynecologic Oncology and Reproductive Medicine, The University of Texas MD Anderson Cancer Center, Houston, TX 77030, USA; scorvigno@mdanderson.org (S.C.); mashaol@mail2.sysu.edu.cn (S.M.); jcelesti@mdanderson.org (J.C.); nfleming@mdanderson.org (N.D.F.); rhajek@mdanderson.org (R.A.H.); alankenau@mdanderson.org (A.L.A.); nbjennin@mdanderson.org (N.B.J.); swestin@mdanderson.org (S.N.W.); aajazaeri@mdanderson.org (A.A.J.); 3Department of Genetics, The University of Texas MD Anderson Cancer Center, Houston, TX 77030, USA; ejthomps@mdanderson.org (E.J.T.); hltang@mdanderson.org (H.T.)

**Keywords:** extracellular vesicle, high-grade serous ovarian cancer, whole-genome sequencing, RNA sequencing, chemotherapy response

## Abstract

**Simple Summary:**

Ovarian cancer is the deadliest gynecological malignancy worldwide, and the biggest issue faced by patients is disease relapse after primary treatment. Understanding up front how patients will respond to therapy is an important and compelling challenge. On these premises, our goal was to understand if a simple blood sample can give us this information, before the patients start chemotherapy. We determined that circulating extracellular vesicles isolated at diagnosis can distinguish between patients who will respond differently to the treatment. These results, if corroborated, will pave the way for a novel, much needed predictive marker of response.

**Abstract:**

Patients with high-grade serous ovarian cancer (HGSC) who have no visible residual disease (R0) after primary surgery have the best clinical outcomes, followed by patients who undergo neoadjuvant chemotherapy (NACT) and have a response enabling interval cytoreductive surgery. Clinically useful biomarkers for predicting these outcomes are still lacking. Extracellular vesicles (EVs) have been recognized as liquid biopsy-based biomarkers for early cancer detection and disease surveillance in other disease settings. In this study, we performed extensive molecular characterization of serum-derived EVs and correlated the findings with therapeutic outcomes in patients with HGSC. Using EV-DNA whole-genome sequencing and EV-RNA sequencing, we identified distinct somatic EV-DNA alterations in cancer-hallmark genes and in ovarian cancer genes, as well as significantly altered oncogenic pathways between the R0 group and NACT groups. We also found significantly altered EV-RNA transcriptomic variations and enriched pathways between the groups. Taken together, our data suggest that the molecular characteristics of EVs could enable prediction of patients with HGSC who could undergo R0 surgery or respond to chemotherapy.

## 1. Introduction

High-grade serous ovarian cancer (HGSC) is a deadly malignancy that lacks reliable biomarkers for diagnosis and prediction of chemotherapy response [1,2]. Patients with HGSC with no visible residual disease after primary surgery (complete gross resection, R0) have the best clinical outcomes, followed by patients who undergo neoadjuvant chemotherapy (NACT) and have a R0 resection at the interval surgery; the worse outcomes tend to occur in patients with residual disease at interval tumor reductive surgery [3,4]. However, the genomic features that best predict R0 surgery and drug response remain unclear, and practical methods for identifying such predictive biomarkers in clinical settings are lacking.

Through comprehensive multi-omics approaches, we have recently discovered significant distinct molecular abnormalities, cellular differences, and immune cell repertoire alterations between highly clinically defined groups of patients who had R0 resection versus those triaged to NACT [5,6]. Our results in tumor specimens point to important underlying biological differences in HGSCs considered resectable versus non-resectable, and in those that respond versus do not respond to NACT. However, identifying these molecular abnormalities and cellular changes requires a tumor tissue specimen. Thus, here, we explore options for testing a more practical liquid biopsy-based approach.

Late diagnosis at advanced stages and a high rate of recurrence after primary therapy represent major factors affecting poor prognosis, which makes compelling the identification of markers for early diagnosis and for patient stratification at primary assessment. Importantly, liquid biopsies have recently emerged as an effective and cost-efficient alternative or companion to the classic radiological and histological assessments in the diagnosis and monitoring of several cancer types. A series of tumor-derived liquid biopsy markers have been identified, and among them, extracellular vesicles (EVs) are quickly becoming one of the most prominent [7,8,9]. These membrane-coated vesicles are formed intracellularly and secreted extracellularly [7], and are generally distinguished according to their size as small EVs (also called exosomes, 50 to 150 nm in diameter), medium EVs (150 to 300 nm in diameter), and large EVs (300 to 1000 nm in diameter) [8]. EVs have brought enthusiasm to the field of translational biomarker research because of their potential for early detection and disease surveillance. EVs, which can be detected in serum, plasma, and ascites, carry molecules such as oncoproteins and oncopeptides; RNA species such as microRNAs (miRNAs), mRNAs, and long non-coding RNAs (ncRNAs); lipids; and DNA fragments. EVs have crucial roles in cancer development and cell-to-cell communication [9].

Despite the existence of FDA-approved tests that detect other liquid-biopsy components (such as circulating cancer cells) for use as diagnostic companions, these applications are often biased by the low specificity of the assays and by the potential confounding factor represented by hematopoietic precursors with clonal mutations [10]. Therefore, EVs represent a novel source of cancer-derived material, whose abundance and relative cancer specificity make them particularly attractive as diagnostic and prognostic markers. The purpose of the current study was to perform extensive molecular characterization of serum-derived EVs and correlate them with therapeutic outcomes in patients with HGSC. We identified distinct EV-DNA genomic and EV-RNA transcriptomic variations, as well as oncogenic pathways in whole-genome sequencing (WGS) and RNA-sequencing (RNA-seq) analyses of serum-derived EVs from clinically well-defined groups of patients with HGSC.

## 2. Materials and Methods

### 2.1. Patient Sample Collection

Blood samples for 24 patients treated under a systematic surgical algorithm at The University of Texas MD Anderson Cancer Center (Houston, TX, USA) were obtained from the Gynecologic Tumor Bank. The cohort considered for this study was composed of patients with advanced stage ovarian cancer who underwent a laparoscopic assessment to assess primary resectability; the algorithm and patient eligibility were previously described [5,6]. Patients had provided written informed consent under an approved protocol by the Institutional Review Board (LAB10-0850). Patients were classified into three groups based on treatment and response: primary surgery with complete gross resection (R0, *n* = 9); NACT (carboplatin and paclitaxel) with excellent tumor response, which was defined as optimal interval debulking surgery, with complete response according to the response evaluation criteria in solid tumors (RECIST 1.1) or only microscopic disease left at the time of interval surgery and/or pathology from interval surgery (NACT-ER, *n* = 8); or NACT with poor tumor response, defined as stable or progressive disease according to RECIST criteria and/or suboptimal interval cytoreduction after 3–4 cycles of NACT (NACT-PR, *n* = 7). Patient clinicopathological characteristics are reported in Appendix A.

### 2.2. Medium and Large EV Isolation from Serum

Serum-derived medium/large EVs (m/lEVs) from patient blood samples, obtained before treatment, were isolated following standard methods according to the International Society for Extracellular Vesicles [11]. In brief, 9 mL of PBS was added to 1 mL of serum per patient and lightly shaken overnight at 4 °C. After a first centrifugation at 300 relative centrifugal force (rcf) for 10 min, the supernatant was transferred to a new tube and subsequently spun at 2000 rcf for 20 min. The supernatant underwent ultracentrifugation at 10,000 rcf for 40 min, and the pellet, composed of m/lEVs, was collected. A final ultracentrifugation step at 10,000 rcf for 40 min was used to wash the pellet, which was then stored at –80 °C until DNA and RNA isolation was performed.

### 2.3. Medium and Large EV Characterization and Quantification

The patient-derived m/lEVs were characterized by both nanoparticle tracking analysis (NTA) and transmission electron microscopy (TEM) (Appendix A). Isolated EV fractions were diluted and analyzed using NTA, to record their number and size. TEM was performed at MD Anderson’s High-Resolution Electron Microscopy Facility and was described previously [11,12]. In brief, formvar/carbon-coated mesh nickel grids were treated with a poly-L-lysine solution for 5 min, and excess solution was blotted with filter paper, to allow the grids to dry. The fixed samples were then added onto the grids, to allow small EVs to absorb for 1 h. Grids were rinsed with several drops of PBS five times, 3 min each, and incubated with 2% glutaraldehyde in PBS for 15 min. Grids were then rinsed with distilled water and incubated with 1% uranyl acetate, and they were allowed to dry before TEM examination. Samples were examined under a JEM 1010 transmission electron microscope (JEOL USA, Peabody, MA, USA) at an accelerating voltage of 80 kV.

### 2.4. Nucleic Acid Isolation from m/lEVs

EV-DNA was isolated using an XCF exosomal DNA isolation kit (catalogue no. XCF200A-1, System Biosciences, Palo Alto, CA, USA), and EV-RNA was isolated from the same patients using ExoRNeasy Midi and Maxi kits (catalogue no. 77023, Qiagen, Hilden, Germany), according to the manufacturer’s instructions. Samples were quantified and sent for analysis.

### 2.5. Whole-Genome Sequencing

Exosome whole-genome sequencing (WGS) was performed at the Advanced Technology Genomics Core at MD Anderson Cancer Center. Illumina libraries were prepared from 1 ng–200 ng of Diagenode Bioruptor Pico sheared exosome DNA, using a KAPA Hyper Library Preparation Kit (KAPA Biosystems, Inc., Wilmington, MA, USA). DNA fragmentation was performed. Libraries were amplified by eight cycles of PCR and assessed for size distribution (4200 TapeStation platform and High Sensitivity D1000 ScreenTape assay from Agilent Technologies, Santa Clara, CA, USA). Quantification of libraries was performed using a Qubit dsDNA HS Assay Kit (Thermo Fisher Scientific, Waltham, MA, USA). Multiplexing of equimolar quantities of the indexed libraries was performed, with four libraries per pool. A qPCR using a KAPA Library Quantification Kit (KAPA Biosystems, Wilmington, MA, USA) was used to quantify the pool, which was then sequenced on a NovaSeq 6000 SP Flow Cell (Illumina, San Diego, CA, USA) using the 100-nt paired-end format. The quality of extracted EV DNA was assessed using a High Sensitivity DNA TapeStation and quantified using a Qubit 2.0 RNA HS assay. DNA was sheared with an LE220 ultrasonicator (Covaris, Woburn, MA, USA), and a library was generated with a NEBNext Ultra II DNA Library Prep Kit for Illumina (New England Biolabs, Ipswich, MA, USA), following the manufacturer’s instructions. The quantity of final libraries was determined using Qubit 2.0; the quality was assessed using the TapeStation D1000/ScreenTape. The size of the final library was about 430 bp, with an insert size of about 300 bp. Equimolar pooling of libraries was performed based on QC (quality control) values, and the libraries were sequenced on an Illumina NovaSeq S4 (read-length configuration of 150 paired ends for 1B paired-end reads per sample). Pair-end sequencing reads in FASTQ format were generated from BCL raw data using Illumina CASAVA software (v1.8.2). The reads were aligned to the hg19 human reference genome using the BWA tool (v0.7.5) [13]. The duplicate reads were removed using Picard tools (unpublished, http://broadinstitute.github.io/picard/ accessed on 18 November 2021), and local realignments were performed using GATK software (v4.1.1.0) [14]. The BAM files were then used for downstream analysis.

### 2.6. Mutation Calling and CNV Identification

The MuTect method (v1.1.4) [15] was used to identify somatic point variants, and the Pindel program (v0.2.4) [16] was used to identify somatic insertions and deletions. To eliminate common polymorphisms, we generated common normal sequence using in-house pooled normal sequence. A series of post-calling filtering was applied for somatic mutations: (a) variant allele frequency (VAF) ≥0.05 in the EV sample and ≤0.02 in the common normal sample, and (b) a population frequency threshold of 0.5%, to filter out common variants in the databases dbSNP129 [17], 1000 Genome Project [18], Exome Aggregation Consortium [19], and ESP6500 [20]. To understand the potential functional consequences of the detected variants, we annotated them using the Annovar tool [21] and the dbNSFP database [22] and compared them with dbSNP [23], ClinVar [24], COSMIC [25], and The Cancer Genome Atlas databases. Copy number variations (CNVs) were identified using the package HMMcopy [26]. The copy number log2 ratios of EV-DNA versus control were calculated across the entire capture regions and then subjected to segmentation using the Circular Binary Segmentation (CBS) method [27]. A cutoff of log2 ratio ≤−0.5 was applied to identify copy losses, and a log2 ratio ≥0.5 was applied for copy gains.

### 2.7. RNA Sequencing

Illumina-compatible low-input total RNA libraries were prepared using a SMARTer Stranded Total RNA-Seq Kit v2—Pico Input Mammalian (Takara Bio, San Jose, CA, USA) at the MD Anderson Cancer Genomics Core Laboratory. Briefly, 2–10 ng of total RNA was converted to cDNA using Takara’s SMART (Switching Mechanism at 5′ End of RNA Template) technology. Illumina adapters with specific barcodes were then added to the ds (double-stranded) cDNA using 12 cycles of PCR. The PCR products were purified using AmPure Beads (Beckman Coulter, Brea, CA, USA), and then ribosomal cDNA was depleted. The cDNA fragments were further enriched by 14 cycles of PCR using Illumina universal library primers, then purified to yield the final cDNA library. Libraries were pooled, four per pool, then sequenced on a NextSeq500 sequencer (Illumina) using the mid-output 150-cycle flow cell with the 75-nt paired-end format. The NGS library was built with SMARter Low-Input Total RNA-Seq. Raw sequencing data were converted to FASTQ files and aligned to the reference genome (hg19) using the Spliced Transcripts Alignment to a Reference (STAR) algorithm [28]. The HTSeq-count package was then utilized to generate raw read counts for each gene [29]. The DESeq2 package was used for data processing, normalization, and differential expression analysis following standard procedures [30]. The differentially expressed genes were selected by the criterion of log2 fold change (L2FC) ≤−1 or ≥1, and the cutoff for adjusted *p*-value was 0.05. The GSEA tool [31] was used for gene set enrichment analysis.

### 2.8. Statistical Analysis

As we previously described [6], to evaluate the statistical significance, Student’s *t*-test was used if the data fit a normal distribution. Otherwise, a Wilcoxon rank-sum test was used to test the differences between groups. The Benjamini–Hochberg method was used to control the false discovery rate. In transcriptomic analysis, the differential expression was evaluated using a negative binomial generalized linear model, as described by DESeq2, and logarithmic fold change and adjusted *p*-values were used for differentially expressed gene assessment.

## 3. Results

### 3.1. Patient Groups and Sequencing Data

The patient cohort and groups were previously reported [5,6]. In brief, a total of 24 patients were included in this study: nine patients who had no visible residual disease after primary surgery (R0), eight who had an excellent response to NACT (NACT-ER), and seven who had a poor response to NACT (NACT-PR) (Appendix A). For each patient, the isolated EV-DNA and EV-RNA were subjected to WGS and RNA-seq, respectively. The average coverage for WGS of EV-DNA was 10×, and the average coverage for RNA-seq of EV-RNA was 15×.

### 3.2. Somatic Alterations in EV-DNA

Somatic variants and CNVs were identified by comparing the EV-DNA versus a pooled common-normal sample. An average of 153 nonsynonymous somatic variants were identified from each sample. Based on a known list of ovarian cancer driver genes and cancer-hallmark genes (https://cancer.sanger.ac.uk/census#cl_sub_tables accessed on 6 December 2021), the somatic alteration landscape was plotted (Figure 1). Overall, seven ovarian cancer-associated genes were found to be altered among nine (38%) of the 24 EV-DNA samples (Figure 1A): *TP53* (4, 17%), *FAT3* (2, 8%), *MLH1* (2, 8%), *ARID1A* (1, 4%), *JAG2* (1, 4%), *NOTCH3* (1, 4%), and *PTEN* (1, 4%). Figure 1B shows the altered cancer-hallmark genes; alterations were seen in 22 (92%) of the 24 samples. The most frequently altered genes were *KMT2C* (6, 25%), *HNRNPA2B1* (4, 17%), *NOTCH1* (4, 17%), and *TP53* (4, 17%). The oncogenic pathway analysis (Figure 1C) further showed that the most frequently altered pathways were the RTK-RAS pathway (16, 67%), Hippo pathway (13, 54%), WNT pathway (13, 54%), and NOTCH pathway (12, 24%).

Figure 2 shows the details of the most frequently altered oncogenic pathways. As shown in Figure 2A, 21 genes in the RTK-RAS pathway were altered, including *ERF* (4, 17%), *ABL1* (2, 8%), and *DAB2IP* (2, 8%). In the Hippo pathway (Figure 2B), 14 genes were altered, with *FAT1* as the most common (3, 13%), as well as *DCHS1* (2, 8%) and *DCHS2* (2, 8%). In the WNT pathway (Figure 2C), 10 genes were altered, including *TLE3* (5, 21%), *APC* (3, 13%), and *FRAT1* (3, 13%). In the NOTCH pathway (Figure 2D), 16 genes were altered, including *PSENEN* (5, 21%), *NOTCH1* (4, 17%), and *NCOR2* (2, 8%). When comparing patient groups, no statistically significant differences for variants and pathways were found between the R0 and the NACT groups, nor between NACT-ER and NACT-PR.

### 3.3. Comparison between EV-DNA and Tumor DNA

We compared the somatic genomic alterations in EV-DNA and tumor DNA from the same patients and found no overlaps of mutations and CNVs. We further investigated the genomic alteration enrichment in ovarian cancer genes and cancer-hallmark genes in EV-DNA compared with tumor DNA. As shown in Appendix A, 14 ovarian cancer genes were found to be enriched with alterations in tumor DNA, including *TP53* (*p* < 0.0001, adj-*p* < 0.0001), *PIK3CA* (*p* < 0.0001, adj-*p* = 0.008), *CSMD3* (*p* = 0.0002, adj-*p* = 0.01), *CCND2* (*p* = 0.0007, adj-*p* = 0.03), *ERBB2* (*p* = 0.003, adj-*p* = 0.06), and *CCNE1* (*p* = 0.003, adj-*p* = 0.06), and no ovarian cancer genes were enriched with alterations in EV-DNA. It is interesting to note that among the cancer-hallmark genes (Appendix A), two genes, *H3F3B* (*p* < 0.0001, adj-*p* < 0.0001) and *B2M* (*p* = 0.0003, adj-*p* = 0.018), were enriched with alterations in EV-DNA, while 55 genes were enriched with alterations in tumor DNA.

### 3.4. Transcriptomic Expression Profiles in EV-RNA

Next, we investigated the gene expression patterns in EV-RNA samples. Overall, a total of 28,246 genes were found to be expressed in the EV-RNA samples. Sample-to-sample distance was calculated, and the patient samples were clustered, as shown in Figure 3A. Samples from the same groups were relatively clustered together, indicating that the EV-RNA expression profiles were more similar within each group than across different groups. Unsupervised clustering using the top 10,000 most variable genes (Figure 3B) identified two main clusters, of which one cluster only included samples from the NACT-ER and NACT-PR groups, while samples from the R0 group were enriched in the other cluster. This suggests distinct expression patterns of the R0 group compared to the NACT groups.

### 3.5. Differential Expression and Enriched Pathways between the R0 and NACT Groups

We identified 574 differentially expressed genes (DEGs) with an adj-*p* <0.05 and absolute L2FC ≥1 in the R0 compared with the NACT groups (Figure 4A,B). Among these, 18 genes were upregulated in the R0 group, and 556 genes were upregulated in the NACT group. Interestingly, six cancer-hallmark genes were found to be significantly downregulated in the EV-RNA samples from the R0 group (Figure 4C): *ASPSCR1* (L2FC = −1.3, adj-*p* = 0.004), *CIC* (L2FC = −1.0, adj-*p* = 0.006), *BCL9L* (L2FC = −1.1, adj-*p* = 0.005), *NOTCH1* (L2FC = −1.1, adj-*p* = 0.01), *FGFR3* (L2FC = −1.2, adj-*p* = 0.004), and *JAG2* (L2FC = −1.2, adj-*p* = 0.005).

Gene set enrichment analysis (GSEA) was performed, to evaluate the enriched pathways between the R0 and the NACT groups. As shown in Figure 4D, 15 pathways were significantly enriched in the NACT group, and the ones with the most significant normalized enrichment scores (NESs) were hallmark myogenesis (NES = −2.5, adj-*p* < 0.0001), hallmark KRAS signaling DN (downregulated) (NES = −2.2, adj-*p* < 0.0001), hallmark apical junction (NES = −2.1, adj-*p* < 0.0001), and hallmark WNT/beta-catenin signaling (NES = −2.1, adj-*p* = 0.0003). In contrast, one pathway was found to be enriched in the R0 group: hallmark UV response DN (NES = 1.5, adj-*p* = 0.02).

### 3.6. Differential Expression and Enriched Pathways between the NACT-ER and NACT-PR

#### Groups

We further compared the EV-RNA expression between the NACT-ER and NACT-PR groups; however, no DEGs were found (Appendix A). GSEA analysis identified eight significantly enriched pathways (Appendix A). Six pathways were significantly enriched in the NACT-ER group: hallmark MYC targets V1 (NES = 2.2, adj-*p* < 0.0001), hallmark unfolded protein response (NES = 1.9, adj-*p* = 0.001), hallmark MTORC1 signaling (NES = 1.8, adj-*p* = 0.0001), hallmark oxidative phosphorylation (NES = 1.7, adj-*p* = 0.0004), hallmark MYC targets V2 (NES = 1.7, adj-*p* = 0.02), and hallmark interferon-gamma response (NES = 1.5, adj-*p* = 0.006). In contrast, two pathways were significantly enriched in the NACT-PR group: hallmark apical junction (NES = −1.7, adj-*p* = 0.002) and hallmark KRAS signaling DN (NES = −1.5, adj-*p* = 0.01).

### 3.7. Comparison between EV-RNA and Tumor RNA

EV-RNA and tumor RNA were compared from the same patient cohort. The PCA (principal component analysis) plot (Appendix A) shows a clear separation between EV-RNA samples and tumor RNA samples, regardless of the response groups, suggesting a distinct transcriptomic pattern between EVs and tumor tissues. Compared to EV-RNA samples, the tumor RNA samples were more widely distributed on the PCA plot, indicating profound heterogeneity among tumor tissue samples. Furthermore, the pathway analysis (Appendix A) shows that the most enriched pathways in EV-RNA samples were hallmark MYC targets V2 and hallmark inflammatory response. The most enriched pathways in the tumor RNA samples were hallmark hedgehog signaling and hallmark TGF-beta signaling.

## 4. Discussion

In the present study, we identified distinct EV genomic and transcriptomic alterations from clinically well-defined HGSC patient groups. We found several oncogenic pathways significantly altered in WGS of the EV-DNA from all 24 samples analyzed. Through RNA-seq, we found 18 genes upregulated in the R0 group and 556 genes upregulated in the NACT group, which mostly mapped to KRAS signaling and hallmark WNT/beta-catenin signaling. Moreover, among the significantly enriched pathways in patients with excellent response to NACT, we identified pathways connected to metabolism and ATP production, such as oxidative phosphorylation, and to immune activation, such as interferon-gamma response.

EV genomic analysis has been performed mostly for detection of cancer-derived mutations and protein alterations, as possible diagnostic companions. In most cancer types, such as prostate cancer [32], pancreatic cancer [33], and glioblastoma [34], EV-DNA analysis has been tested for cancer detection more than for prognosis and treatment prediction. When a prognostic value was attributed to EV-DNA in previous studies, end-points such as progression-free survival were chosen for the analysis [35]. In ovarian cancer, this is one of the first reported studies in which a comprehensive genomic analysis was applied to circulating EVs in patients stratified according to response to treatment. Here, we focused on a specific cohort of patients undergoing standard surgical and chemotherapy treatment whose surgical resection and NACT outcomes were evaluated according to strict and well-established criteria [3]. To answer the question of whether liquid-biopsy markers might help to predict treatment outcome, our findings demonstrate that there are substantial differences in the circulating EV-RNA of patients who underwent R0 tumor debulking versus patients stratified to receive NACT. Many studies evaluating EVs as predictive biomarkers focused only on detection of single genes [36,37,38], while we applied a more comprehensive transcriptomic analysis. Nonetheless, due to the highly standardized criteria, our study was limited by the relatively small sample size, which calls for validation in bigger cohorts.

EV-DNA sequencing showed that not all samples displayed alteration of cancer-associated genes; in particular, *TP53* was found to be altered in only 4 of the 24 EV-DNA samples. Instead, when oncogenic pathway analysis was performed, the pathway altered throughout the majority of samples (16 of 24) was the RTK-RAS pathway. Interestingly, *RAS* mutation—and in general, *RAS* activity—seems to be involved in the regulation of EV biogenesis rate in cancer cells [39,40]. Similarly, the WNT pathway was among the most frequently altered. EV-mediated delivery of Wnt ligands is known to induce migratory and proliferative potential in recipient cells [41,42,43]. Despite the absence of overlap between somatic alterations in EV-DNA and matched tumor-tissue DNA in our cohort, we believe that circulating EV-DNA still reflects the complex genomic landscape of the tumor microenvironment. EVs’ cargo might not reflect the original cell content but might be regulated by the activation or overexpression of certain genes in the cell of origin. For example, activation of epidermal growth factor receptor (EGFR) signaling by prostate cancer cells in vitro has been associated with the formation of a specific subtype of larger EVs (up to 10 µM in diameter) [31], while in glioblastoma cells, overexpression of EGFR vIII promoted release of EVs enriched with focal adhesion and pro-invasive proteins [32]. Therefore, the clonal diversity present in the tumors might be reflected in the EVs’ composition and cargo, and not necessarily in the overlapping of their genome with the cell of origin. Our analysis of the EV transcriptomic variations instead revealed specific alterations in gene expression directly related to the treatment outcome. The R0 and the NACT groups independently clustered according to gene expression in two distinct groups. Among the specific cancer genes downregulated in the EV-RNA from the R0 group, *NOTCH1* stands out for its specific role in ovarian cancer chemoresistance [44]. Among the enriched pathways in the NACT group, KRAS signaling and WNT/beta-catenin signaling are notably connected to oncogenic potential and tumor progression [45,46].

It is noteworthy that the comparison between tumor tissue DNA and EV-DNA did not report any overlap between the patterns of mutation; similarly, the comparison between EV-RNA and tumor RNA showed a quite different transcriptomic pattern, almost suggesting that the nucleic acids incorporated into EVs are not generated by a casual segregation, but by a well-defined loading program.

## 5. Conclusions

In-depth analysis of liquid biopsy-based circulating markers represents the most accurate approach to identifying specific biomarkers of response in ovarian cancer; in line with that, our finding of a significant variation between gene expression in EV-RNA from patients with different responses to primary treatment may have deep therapeutic and prognostic implications for patients with HGSC.

## Figures and Tables

**Figure 1 cancers-14-03589-f001:**
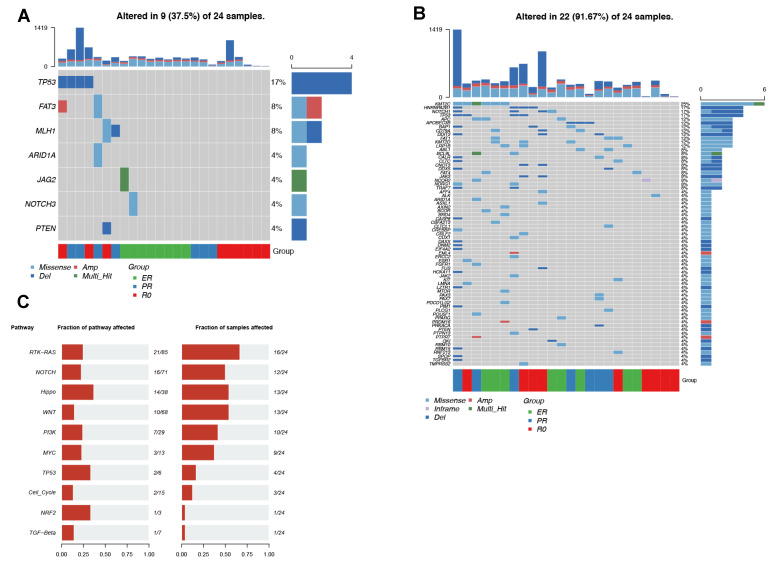
Somatic alterations were identified in 24 EV-DNA samples. The plots show the presence of alterations in cancer-associated and cancer-hallmark genes, and the most frequently altered pathways. (**A**) Oncoplot showing the frequency of each type of somatic alteration in ovarian cancer genes. Each column represents one sample. Patient groups are represented by the colored bars at the bottom, labeled “Group.” (**B**) Oncoplot showing the somatic alterations identified in cancer-hallmark genes. (**C**) Barplots showing the altered oncogenic pathways. Left: The fraction of genes altered in the pathways. Right: The fraction of EV-DNA samples carrying alterations in the pathways. ER = excellent response to neoadjuvant chemotherapy (NACT); PR = poor response to NACT; R0 = no residual disease; EV = extracellular vesicles.

**Figure 2 cancers-14-03589-f002:**
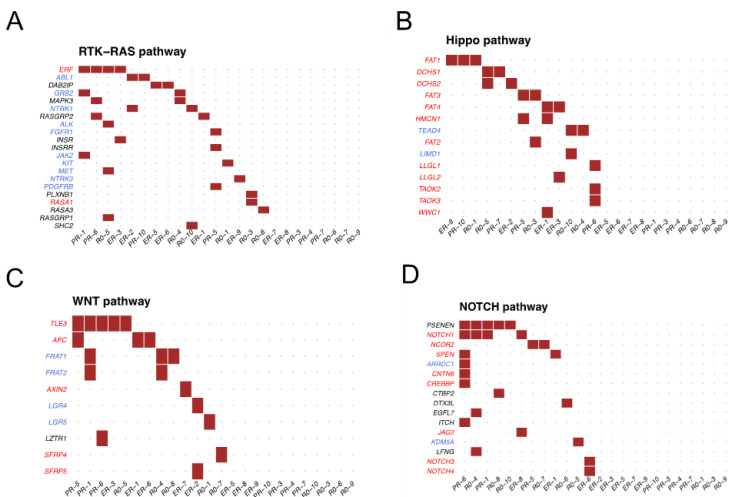
Altered genes for each patient in the four most frequently altered pathways. (**A**) RTK-RAS pathway, (**B**) Hippo pathway, (**C**) WNT pathway, (**D**) NOTCH pathway. Tumor suppressor genes are in red, and oncogenes are in blue.

**Figure 3 cancers-14-03589-f003:**
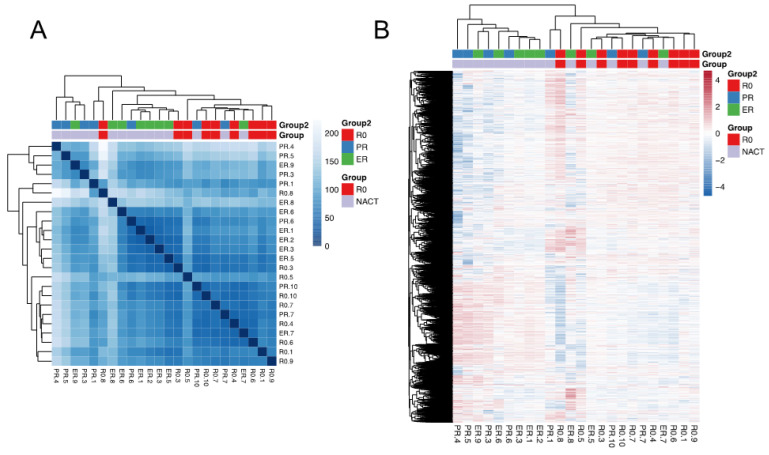
Sequencing analysis for 24 EV-RNA samples. Clustering of samples according to EV-RNA expression is shown; samples tend to cluster according to R0 and NACT groups. (**A**) Heatmap of sample-to-sample distance. (**B**) Unsupervised clustering of the top 10,000 most variable genes. ER = excellent response to neoadjuvant chemotherapy (NACT); PR = poor response to NACT; R0 = no residual disease; EV = extracellular vesicles.

**Figure 4 cancers-14-03589-f004:**
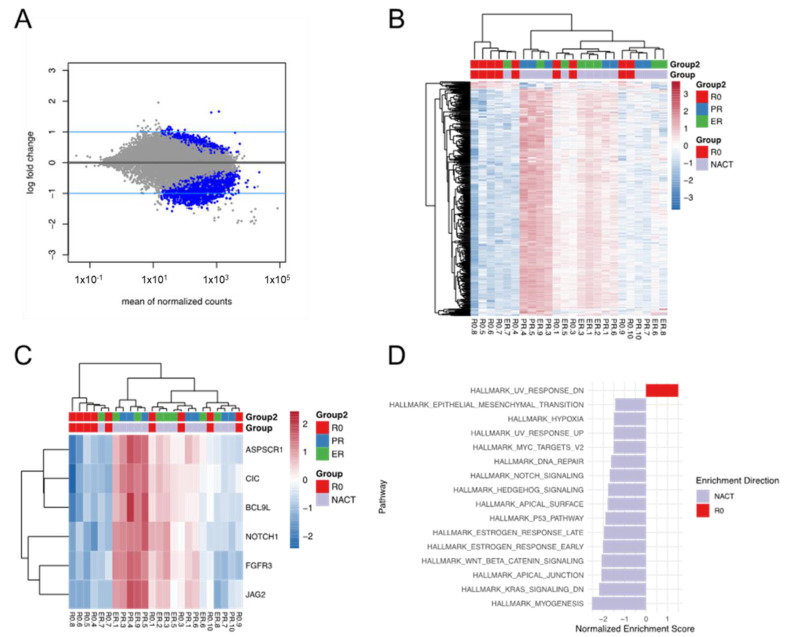
Differential expression analysis of EV-RNA in the R0 vs. NACT groups: The plots show differentially expressed genes between patients who had R0 tumor reductive surgery upfront and those who had neoadjuvant chemotherapy (NACT), with six cancer-hallmark genes found to be significantly downregulated in the R0 group. (**A**) MA plot showing the identified genes, with the blue lines representing the cutoffs for differentially expressed genes (DEGs) between the R0 group and the combined NACT groups. The absolute value of L2FC ≥1 was used as the cutoff for DEGs. The blue dots represent the genes with adjusted *p*-values (adj-*p*) <0.05. (**B**) Heatmap of 547 identified DEGs. (**C**) Heatmap of differentially expressed cancer-hallmark genes. (**D**) GSEA analysis shows the enriched cancer-hallmark pathways in the R0 vs. NACT groups. ER = excellent response; PR = poor response; R0 = radical surgery.

## Data Availability

The WGS and RNA-seq data in this study have been deposited in the European Genome-phenome Archive. Study ID: EGAS00001006350 (Dataset ID for WGS: EGAD00001008973; for RNA-seq: EGAD00001008972).

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
