# Peer review of "Molecular Profiles of Serum-Derived Extracellular Vesicles in High-Grade Serous Ovarian Cancer"

_cancers, 2022, doi:10.3390/cancers14153589_

Round 1

Reviewer 1 Report

My dears,

First of all congratulations for your work. I quite enjoyed reading it.

Please find some comments in the attached pdf file. I would also like to mention the following:

- do you happen to have some follow up of these patients? so that you can show a progression free survival curve or overall survival curves?

- when you classify the NACT ER and NACT PR I suggest explaining these in terms of RECIST criteria. I assume that NACT ER will be complete responders while NACT PR are all the rest.

- if you manage I suggest trying to find some GEO datasets of OV samples similar to yours, to check for the same genes/pathways that you report here. If you find similar results in other contexts it could be a good external validation of your findings.

Looking fw to your revised manuscript!

Best of luck!

Author Response

Reviewer #1:

First of all congratulations for your work. I quite enjoyed reading it.

Response: We thank the reviewer for the positive comments.

Q1: Do you happen to have some follow up of these patients? so that you can show a progression free survival curve or overall survival curves?

Response: We had previously attempted to investigate if there is any correlation between the clinical outcome and each group, and we observed no significant differences. Nevertheless, we retrieved data for PFS (from last day of chemotherapy to date of relapse) in the NACT ER and NACT PR group; the average PFS was 148.3 and 119.3 days, respectively and standard deviation of 155.0 and 70.0 days, respectively.

Q2: when you classify the NACT ER and NACT PR I suggest explaining these in terms of RECIST criteria. I assume that NACT ER will be complete responders while NACT PR are all the rest.

Response: We have now updated the Materials and Methods section and included description of the use of RECIST criteria (page 3, lines 102 to 109).

Q3: if you manage I suggest trying to find some GEO datasets of OV samples similar to yours, to check for the same genes/pathways that you report here. If you find similar results in other contexts it could be a good external validation of your findings.

Response: We appreciate the reivewer’s excellent point and suggestion. We attempted to identify available GEO datasets regarding the characteristics of EV-DNA and EV-RNA in chemotherapy response in high-grade ovarian cancers. However, we could not identify any other publicly available data sets; our manuscript is one of the first reports of comprehensive EV genomic and transcriptomic analyses in highly clinically defined HGSC patient groups.

Comments on the text:

Page 1:

Comment 1: optimal interval cyto surgery. because those that undergo IDS with a residue after NACT have a worse prognosis...

Response: We added in the text “..and have R0 resection at the interval surgery; the worse outcomes tend to occur in patients with residual disease at interval tumor reductive surgery ”, page 2, line 51.

Comment 2: here you mean patients with upfront surgery vs NACT then surgery with whatever outcome in terms on residual disease? I am asking because usually the patients that can undergo upfront surgery are of lower stage so this can be a source of bias for any type of analysis if not accounted for.

Response: The cohort considered for this study was composed of patients with advanced stage ovarian cancer who underwent a systematic laparoscopic surgical algorithm to assess primary resectability. They all presented with advanced disease (Fleming ND et al., 2018). We have clarified this information in theMaterials and Methods section, page 3, lines 98 to 100.

Page 4:

Comment 1: write here that these patients had optimal IDS

Response: We corrected the sentence in the text, page 3, lines 99 and 100, “which was defined as optimal interval debulking surgery, with complete response according to RECIST criteria”.

Comment 2: please add STD DEVIATION next to the mean and also do a ANOVA and chi square tests as appropriate in order to show that the groups are homogenous. Also add a column stating the number of NACT cycles before surgery and CC score

Response: We added standard deviation values to each numerical column and performed ANOVA test. There were no significant differences in age, BMI and CA125 (Brown-Forsythe test not significant for age, BMI, and CA125, Bartletts’s test significant for CA125, p =  0.03). This result has been added to the Supplementary Table 1 legend. All the patients received three to four cycles of NACT; we have now updated “patient sample collection” in the Materials and Methods section. CC score for ER patients is 0 or 1, while it is above 1 for PR patients.

Page 8:

Comment 1: explain the ER PR patient groups

Response: We added a line in the legends for Figures 1, 3, and 4: “ER = excellent response to neoadjuvant chemotherapy (NACT); PR = poor response to NACT; R0 = no gross residual disease”, in page 8 lines 256-258; page 10, lines 304-305; page 11, lines 336-337.

Page 12:

Comment 1: these results should be in the main paper because these pathways are involved in NACT response. This is a really important result (about SF3)

Response: We reported in the main body the finding with the highest significance. We observed no differentially expressed genes (DEGs) between the NACT-ER and NACT-PR groups; this is why we considered the results relative to this comparison, less prominent.

Page 13:

Comment 1: this is for my own curiosity, are there any studies analyzing the origin of the EV? like if they do not have the p53 mutation, is it because they originate from other cells (like tumoral microenvironment)? or is it because EVs are full of other molecules more important than DNA (such as metabolites, miRNAs etc?)

Response: We appreciate this comment. The origin of EVs is not completely understood. Most studies consider that they are formed via endosome maturation which develops into multivesicular bodies and then vesicles. Even if the exact mechanism for cargo uploading is unknown, we and others have demonstrated that DNA-positive EVs are more likely to come from cancer cells (Yokoi et al. 2019). Others have also determined how DNA secretion into EVs might be an active process regulated by specific routes (Jeppesen et al. 2019). Nonetheless, the reason why  EVs with heterogeneous cargo are found in liquid biopsies might be reflective of their origin from diverse cells present in the circulation and in the tumor microenvironment.

Reviewer 2 Report

Reviewer comments:

Comments to the Author

This manuscript targets on the circulating extracellular vesicles isolated at diagnosis could be a probable way to distinguish between patients that will respond differently to the treatments. This article focusses on how a simple blood sample from the patients would give the information how the patient will respond to therapy before the patients start chemotherapy.

This manuscript is impressive in terms of the concept of utilizing circulating vesicles to determine the probable respond to therapies. The manuscript is written well with substantial evidence of data and literature available. The discussion is also well goes with the results and postulated according to the evidence provided.

Minor criticisms

• Authors should increase the cohort of patients to validate their findings.

• Please provide some details about the data in the figure legends.

• Please undergo a thorough check of the manuscript for typographical and grammatical errors.

Author Response

Reviewer #2:

This manuscript targets on the circulating extracellular vesicles isolated at diagnosis could be a probable way to distinguish between patients that will respond differently to the treatments. This article focusses on how a simple blood sample from the patients would give the information how the patient will respond to therapy before the patients start chemotherapy. This manuscript is impressive in terms of the concept of utilizing circulating vesicles to determine the probable respond to therapies. The manuscript is written well with substantial evidence of data and literature available. The discussion is also well goes with the results and postulated according to the evidence provided.

Response: We thank the reviewer for the positive comments.

Q1: Authors should increase the cohort of patients to validate their findings.

Response: We recognize the importance of expanding the cohort of patients. We have acknowledged in the Discussion section the limitation of the current sample size. Moreover, we are expanding our cohort for further validation studies.

Q2: Please provide some details about the data in the figure legends.

Response: Details on the results have been added to the legends for Figures 1, 3, and 4 (page 7 lines 249-251; page 10 lines 301-303; page 11, lines 327 to 330), as requested.

Q3: Please undergo a thorough check of the manuscript for typographical and grammatical errors.

Response: We have carefully reviewed the manuscript again to eliminate any typographical and grammatical errors. Morever, our manuscript was also reviewed by Dr. Patterson in the Editing Services at The University of Texas, MD Anderson Cancer Center.